# Evaluating the Virology and Evolution of Seasonal Human Coronaviruses Associated with the Common Cold in the COVID-19 Era

**DOI:** 10.3390/microorganisms11020445

**Published:** 2023-02-10

**Authors:** Cameron M. Harrison, Jayden M. Doster, Emily H. Landwehr, Nidhi P. Kumar, Ethan J. White, Dia C. Beachboard, Christopher C. Stobart

**Affiliations:** 1Department of Biological Sciences, Butler University, Indianapolis, IN 46208, USA; 2Department of Biology, DeSales University, Central Valley, PA 18034, USA

**Keywords:** coronavirus, common cold, COVID-19

## Abstract

Approximately 15–30% of all cases of the common cold are due to human coronavirus infections. More recently, the emergence of the more severe respiratory coronaviruses, SARS-CoV and MERS-CoV, have highlighted the increased pathogenic potential of emergent coronaviruses. Lastly, the current emergence of SARS-CoV-2 has demonstrated not only the potential for significant disease caused by emerging coronaviruses, but also the capacity of novel coronaviruses to promote pandemic spread. Largely driven by the global response to the COVID-19 pandemic, significant research in coronavirus biology has led to advances in our understanding of these viruses. In this review, we evaluate the virology, emergence, and evolution of the four endemic coronaviruses associated with the common cold, their relationship to pandemic SARS-CoV-2, and discuss the potential for future emergent human coronaviruses.

## 1. Overview of Human Coronaviruses and Their Public Health Impacts

Coronaviruses are zoonotic viruses that are known to infect a wide range of animal hosts. There are seven known human coronaviruses (HCoVs): HCoV-229E, HCoV-NL63, HCoV-OC43, HCoV-HKU1, SARS-CoV, MERS-CoV, and SARS-CoV-2. These human CoVs primarily infect the respiratory system and vary in the severity of disease caused [1]. HCoV-229E, HCoV-OC43, HCoV-NL63, and HCoV-HKU1 typically cause mild diseases, including 15–30% of cases of the common cold [2]. In contrast, the more pathogenic viruses, SARS-CoV, MERS-COV, and SARS-CoV-2, cause severe diseases including COVID-19 and acute respiratory distress syndrome (ARDS).

### 1.1. Human Coronaviruses Associated with the Common Cold

In 1965, the first human coronavirus was identified [3]. The identified isolate, B814, led to the discovery of HCoV-229E (which was found in a standard tissue culture) and HCoV-OC43 (which was isolated from a tracheal organ culture) [2]. Individuals infected with these two strains (HCoV-229E and HCoV-OC43) experienced common symptoms, which included headache, sneezing, cough, fever, malaise, and nasal discharge [4]. These two strains were the only HCoVs identified prior to the 2002 SARS-CoV epidemic. In 2004, HCoV-NL63 was identified in a nasopharyngeal aspirate from a seven-month-old child [5]. HCoV-NL and HCoV-NH were also discovered at the same time, and are likely strains of HCoV-NL63 [1,2,3,4]. While most of the diseases associated with this virus present as a common cold, it is also associated with hypoxia and croup [6]. In 2005, HCoV-HKU1 was first isolated in Hong Kong from an elderly man who was diagnosed with pneumonia [7]. This virus also produced common cold symptoms like HCoV-229E and HCoV-OC43 during infection. All four common cold coronaviruses have since been shown to be globally distributed and to be endemic [2,8,9].

### 1.2. Emergent Coronaviruses—SARS-CoV, MERS-CoV, and SARS-CoV-2

The first highly pathogenic HCoV emerged in November of 2002 in the Guangdong Province of China [10]. Severe acute respiratory syndrome coronavirus (SARS-CoV)-infected individuals displayed symptoms including a fever, myalgia, headache, malaise, dry cough, dyspnea, respiratory distress, and diarrhea [4]. With cases being detected across the province as an atypical pneumonia, the World Health Organization issued a global alert, and the provincial health department instituted a variety of public health measures to limit exposure and contain the virus [10]. Even with these measures taken, 8098 people across 29 countries were reported to have developed SARS-CoV disease with a 9.8% case fatality rate [11]. The public health measures contained the epidemic within a year; however, a laboratory-acquired infection in 2004 was observed [12]. Middle East Respiratory Syndrome Coronavirus (MERS-CoV) emerged in Saudi Arabia in 2012. It has spread to 27 other countries, and has a case fatality rate of approximately 35% [13]. Like other human coronaviruses, MERS-CoV infects the respiratory system and has symptoms that include fever, cough, chills, sore throats, myalgia, pneumonia, diarrhea, and vomiting [4].

In December 2019, SARS-CoV-2 emerged in the Wuhan Province of China. The virus was isolated from the bronchoalveolar lavage fluid of a forty-one-year-old man who was admitted to a hospital in Wuhan [14]. Sequencing revealed a novel coronavirus that was closely related to a SARS-like coronavirus that had been seen in bats. The novel coronavirus was named WH-Human 1 coronavirus (WHCV), which was later changed to 2019-nCoV and finally renamed SARS-CoV-2 [14]. To date (as of December 2022), the WHO has reported at least 652 million confirmed cases and 6.6 million deaths.

## 2. Coronavirus Biology

### 2.1. Classification

According to the International Committee on the Taxonomy of Viruses (ICTV), coronaviruses are currently classified as members of the family *Coronaviridae* and subfamily *Orthocoronavirinae* within the order *Nidovirales*. Coronaviruses share similar virion structures and, along with other nidoviruses, express their genes through 3′-nested subgenomic RNAs [15]. There are currently four genera that are used for the classification of coronaviruses (*Alphacoronavirus*, *Betacoronavirus*, *Gammacoronavirus*, and *Deltacoronavirus*) based on genetic composition and antigenic cross-reactivity [16,17]. Alpha- and betacoronaviruses are largely specific to mammalian hosts, whereas most gamma- and deltacoronaviruses infect birds (and to a lesser extent, mammals) [18]. All seven known HCoV strains recognized to date are either alphacoronaviruses (HCoV-229E and HCoV-NL63) or betacoronaviruses (HCoV-HKU1, HCoV-OC43, SARS-CoV, MERS-CoV, and SARS-CoV-2) [16].

### 2.2. Structure

Coronaviruses are enveloped, positive-sense RNA viruses which express genomes ranging from 27 to 32 kb in size. Coronaviruses were originally proposed as a group of viruses based on their largely spherical or pleiomorphic shape and the distinct surface expression of their spike attachment proteins, as observed by electron microscopy, giving the virus its crown- or corona-like appearance (Figure 1A) [19,20,21]. Coronaviruses share the expression of four structural proteins which comprise their virions and are essential for infectious complete virion formation: envelope (E), matrix (M), nucleocapsid (N), and spike (S) (Figure 1B) [15]. The E protein is a short integral protein which is expressed at high levels within infected cells, is incorporated into the viral envelope membrane, and aids in coronavirus assembly and budding [22,23,24]. The coronavirus M protein coordinates virion assembly through its interactions with all three other structural proteins (S, E, and N) to ensure complete virion assembly; and, along with E, it is found comprising the viral envelope structure [15,25,26,27]. The N protein, unlike the other three structural proteins, is mostly involved internal to the virion and associates with the positive-sense RNA genome comprising the viral nucleocapsid. Studies have also demonstrated a role of the N protein in aiding in assembly, budding, and regulation of viral biosynthesis [28,29].

Lastly, the coronavirus spike glycoprotein is essential for mediating both the attachment and fusion of the virus to host cells. Spikes form trimers, with each monomer consisting of three different structural components: (1) a large ectodomain, which contains both the receptor-binding subunit (S1) and the membrane-fusion subunit (S2); (2) a transmembrane domain for insertion into the viral envelope; and (3) an intracellular tail [30,31,32,33]. During viral entry, the spike glycoprotein undergoes a conformational change from a pre-fusion to a post-fusion form to mediate membrane fusion [32]. Distinct differences in both the structures of different coronavirus spike proteins as well as the mechanisms of activation have been observed among different coronaviruses, which correlate to some degree with their phylogeny [34]. Unsurprisingly, given their common receptor and high protein sequence identity, the spike glycoprotein structures of SARS-CoV and SARS-CoV-2 show high homology. However, there also appears to be some level of similarity between the spike of SARS-CoV-2 and that of the other betacoronaviruses HCoV-HKU1 and HCoV-OC43 [34]. The coronavirus spike protein is responsible for receptor recognition and viral tropism, and structural differences in spike define both immune recognition and potential for immune escape [35]. These studies and others collectively suggest that cross-reactive immune responses may be possible between SARS-CoV-2 and common cold coronaviruses (HCoV-229E, HCoV-NL63, HCoV-HKU1, and HCoV-OC43) due to similarities in the antigenic presentation and function of the coronavirus spike proteins [34,35,36,37,38].

### 2.3. Genomic Organization and Phylogeny of Human Coronavirus Common Cold Strains and SARS-CoV-2

Human coronaviruses have positive-strand RNA genomes, which range in size from 27.3 to 30.7 kb. The genome contains a 5′-cap and a poly-A tail to facilitate translation upon uncoating in host cells. The first approximately two-thirds of the coronavirus genome encodes the large replicase open-reading frames (ORFs)—ORF1a and ORF1ab (Figure 2) [15,39]. The translated replicase polyproteins, pp1a, and via a ribosomal frameshift, pp1ab, contain up to sixteen nonstructural proteins (nsps) that are proteolytically processed by two or three viral encoded proteases (papain-like protease(s), PLP(s), and the 3C-like protease, 3CLpro) [40,41,42]. Among the nsps encoded by the replicase ORF are the viral RNA-dependent RNA polymerase (nsp12), at least two enzymes with proteolytic activity (nsp3 [housing PLP(s)] and nsp5 [3CLpro]), an RNA proofreading enzyme (nsp14), multiple host-modulatory genes, and several cofactors involved in either replication or genomic capping [43,44,45,46]. The remaining third of the genome encodes the four common structural genes (E, N, M, and S), as well as an array of accessory genes which show variability between coronavirus genera [15].

The four human coronaviruses associated with the common cold exhibit several unique differences in their sequences that may account for the observed differences in tropism and pathogenesis among the viruses. Upon its initial identification in 2004, sequence analysis of *Alphacoronavirus* HCoV-NL63 showed that it shares only 63% and 75% identity with the ORF1a and 1b nucleotide regions of HCoV-229E, despite being its closest human relative and sharing the same genus [47]. Phylogenetic analysis suggests that the two viruses diverged approximately 1000 years ago [48]. In addition, HCoV-NL63 contains a unique large insertion in its spike gene, which is absent in the spike of HCoV-229E. It is speculated that this insertion may play a role in the observed differences in host cell receptors, host range, and disease outcomes associated with the two viruses [6,47,49]. Human betacoronaviruses HKU1 and OC43 share a similar genomic organization and over 80% nucleotide sequence identity of their replicase genes; however, several unique differences have also been observed [7]. An acidic tandem repeat in the N-terminus of nsp3 associated with PLP1 has been observed in most HCoV-HKU1 strains to date, but is markedly absent from other human coronaviruses [7,50]. Despite this difference in the structures of their papain-like proteases, the 3CLpro proteases of HKU1 and OC43 have been shown to be functionally compatible [51]. In addition, HCoV-HKU1 exhibits the most extreme codon usage bias and the lowest G+C content (32%) of all known coronaviruses [52,53]. Genetic comparisons between *Alpha*- and *Betacoronavirus* genera show sequence identities markedly drop to below 60% [7]. Recombination is known to regularly occur with coronaviruses and evidence of recombination events has been found in sequence studies for all four common cold strains [54,55,56,57]. Furthermore, these recombination events have led to the formation of different genotypes of the four coronaviruses and evidence of intra-genera recombination events has also been observed, although they are certainly rarer [56].

The emergence of SARS-CoV-2 in 2019 has led to significant advances in our understanding of coronavirus genetics and evolution. SARS-CoV-2 shares a genome-wide sequence identity with SARS-CoV of 79%, with the highest genomic conservation associated with ORF1b and N, and the greatest divergence in ORF1a and spike [58]. Significantly greater similarities have been observed between SARS-CoV-2 and several known bat coronaviruses, which have been implicated in the zoonotic origin of the virus [16,58,59,60]. While SARS-CoV-2 shares classification as a *Betacoronavirus* with both HCoV-HKU1 and HCoV-OC43, the genomic organization and sequence homology remains quite low between the viruses.

### 2.4. Replication

Coronavirus infections begin with attachment at the cell surface of the respiratory epithelium, facilitated by the viral spike attachment protein. The receptors for the coronavirus common cold viruses differ between strains and define the tropism and host range of each virus. HCoV-229E attaches to aminopeptidase N (AP-N; also known as CD13), a glycosylated protease found on an array of both respiratory and enteric epithelial cells [61]. Like AP-N, angiotensin converting enzyme 2 (ACE2), which serves as the receptor for HCoV-NL63, is also found on both respiratory and enteric epithelial cells, as well as several other cell types throughout the human body, where it plays a key role in the renin-angiotensin system [62,63]. Interestingly, ACE2 also serves as the primary receptor for both SARS-CoV and SARS-CoV-2; however, the binding affinity of HCoV-NL63 has been demonstrated to be less than that of SARS-CoV [63]. In contrast to the other human coronaviruses, both HCoV-HKU1 and HCoV-OC43 use a carbohydrate ligand for binding, 9-O-acetylated sialic acid (9-O-Ac-Sia) [64,65]. Coronaviruses differ on whether the N-terminal or C-terminal domain of the S1 subunit of the spike facilitates attachment to the host receptor protein [30]. However, upon engagement either at the cell surface or an endocytic compartment, the S2 subunit of the spike triggers membrane fusion [66]. The mechanisms and dynamics of spike-mediated fusion differ for each coronavirus and are dependent on several different factors, including pH and at least one spike cleavage event [30].

Upon entry, the positive-strand RNA genome of coronaviruses undergoes translation of the replicase ORF, resulting in the production of two polyproteins: pp1a (encoding nonstructural proteins 1–11) and pp1ab (encoding nonstructural proteins 1–16) (Figure 2). The switching of translation between pp1a and pp1ab is driven by the presence of an RNA pseudoknot structure, which causes a -1 ribosomal frameshift [40,67]. These polyproteins undergo co- and post-translational maturation cleavage events mediated by the one or two PLP protease subunits of nsp3 (which are responsible for cleavage events between nsps 1–4), as well as the 3CLpro protease (nsp5) (which cleaves between nsps 4–16) [51,68,69]. During cleavage, the coronavirus nsps begin to assemble into replication-transcription complexes (RTC) to both drive synthesis of the structural and accessory ORFs, as well as genomic replication [70,71]. Several of the replicase nsps, such as nsps 3, 4, and 6, contain hydrophobic transmembrane-spanning regions that promote targeting of the assembling RTC to host intracellular membranes [72]. Collectively, these proteins induce visible cytopathic changes, including the formation of double membrane vesicles (DMV) and convoluted membranes (CV) on host endoplasmic reticulum (ER) and ER-Golgi intermediate compartments (ERGIC), which are hypothesized to aid in the concentration of viral biosynthesis materials, as well as shield viral RNAs from detection by host innate immune factors [72,73,74,75,76].

Coronavirus RTCs are responsible for the synthesis of both genomic RNAs, as well as sub-genomic RNAs that encode the structural and accessory ORF gene products [77]. These subgenomic RNAs are generated as a nested set of mRNAs [39,78]. In the assembly of each molecule, the 5′-untranslated region (UTR) is fused to the transcriptional regulatory sequence (TRS) associated with the downstream ORF through a polymerase template switching before transcribing the remainder of the template RNA complete with a 3′-terminal polyadenylated tail [77,78,79]. The capacities of the coronavirus RdRp to drive strand-switching during RNA synthesis and the viral exoribonuclease enzyme (nsp14) to mediate proofreading are major factors for the documented recombination capacity of the virus [80,81]. After structural gene mRNA synthesis, the structural gene products are translated into the endoplasmic reticulum and trafficked to the ERGIC, where they promote viral assembly with newly formed genomic RNAs that are encapsidated by N proteins [82,83]. Mature fully-formed virions are trafficked to the cell surface in either secretory vesicles or deacidified lysosomal organelles, whereby they undergo secretion via exocytosis [76,84]. For some coronavirus strains, the accumulation of unincorporated spike proteins on the cell surface may induce cell-to-cell fusion and the formation of multinucleated syncytia [85]. This process aids the intercellular spread of the virus without requiring virion release.

## 3. Coronavirus Pathogenesis and Disease

### 3.1. Human Coronavirus Diseases

Human coronavirus disease falls into two categories: (1) common-cold-like diseases caused by HCoV-229E, HCoV-NL63, HCoV-HKU1, and HCoV-OC43, and (2) lower respiratory tract diseases associated with the more pathogenic SARS-CoV, MERS-CoV, and SARS-CoV-2. The severity of diseases for all HCoVs seems to depend on age, immune status, and co-morbidities within the individual. The HCoVs associated with the common cold infect the upper respiratory tract and cause symptoms including headaches, rhinorrhea, malaise, and sore throats [4]. Both SARS-CoV and MERS-CoV are associated with lower respiratory disease and often result in acute respiratory distress syndrome (ARDS). Interestingly, depending on the individual, SARS-CoV-2 has a spectrum of disease that ranges from asymptomatic, to mild diseases (with the cold-like symptoms described above) of the upper respiratory tract, to more severe diseases of the lower respiratory tract [86]. In common with SARS-CoV-2, the common-cold-associated HCoVs can also cause more severe diseases. The common cold viruses can lead to lower respiratory tract infections and progress to more severe diseases in some instances. For instance, HKU1 has been shown to be neuroinvasive and cause a high rate of febrile seizures [87,88]. HCoV-229E and HCoV-OC43 have also been shown to have neurotropism [89,90]. Additionally, NL-63 is associated with hypoxia and croup in young children [6].

### 3.2. Transmission and Cellular Infection

Human-to-human transmission for all HCoVs involves spread by respiratory droplets from coughing or sneezing, as well as by fomites [2,4]. Fomites act as a substantial reservoir of disease that can be easily transferred from surface contact to the eyes, nose, or mouth. The persistence of HCoV-229 on surfaces has been tested and the virus remained infectious for at least 5 days [91].

During the SARS-CoV epidemic, public health measures taken included quarantining when symptoms appeared, the cancellation of mass gatherings, and controlling the borders of viral hotspots. This included testing travelers and travel advisories. Additionally, there were recommendations of wearing masks and re-education of the population about the importance of personal hygiene (washing hands and appropriate technique for sneezing/coughing) [10,12]. These measures were able to limit and contain the SARS-CoV epidemic and continued to do so when another smaller outbreak occurred in 2004 [12]. When SARS-CoV-2 emerged, the same measures were put into place; however, key differences in when the virus is transmissible (prior to onset of symptoms with SARS-CoV-2) meant that more stringent measures had to be implemented. These included cancellation of mass gatherings, quarantining not just individuals with symptoms but others through contact tracing, travel advisories, and widespread testing. While these measures were beneficial, they have not prevented, but only mitigated, the spread of SARS-CoV-2 [92,93,94].

One of the factors that can contribute to the tropism and pathogenicity of HCoVs is the interaction of spike proteins with the host cell receptors and proteases that mediate cleavage and exposure of the fusion peptide. It is suggested that the receptor and protease determine the tropism and spread within the host [89]. A spike protein can be cleaved between the S1 (receptor binding domain) and the S2 (fusion peptide), or it can be cleaved at a site adjacent to the S2 (S2′). Using HCoV-OC43, it was demonstrated that S2′ cleavage by endosomal proteases is important for efficient entry and infection of neuronal cells [89]. It remains to be determined whether other neuroinvasive viruses exhibit the same usage of endosomal proteases to cleave the S2′ site on the spike protein. SARS-CoV, SARS-CoV-2, and HCoV-NL63 all use ACE2 as a host receptor. While this receptor is broadly distributed within the human body, it may not be the only determinant for viral spread [95]. For example, it is rare for common cold viruses like HCoV-NL63 to be associated with gastroenteritis, which is often found in enterovirus co-infections [96]. However, gastroenteritis is seen in SARS-CoV, MERS-CoV, and SARS-CoV-2 [4]. It is possible that protease cleavage of Spike in other tissues may be responsible for this discrepancy.

### 3.3. Modulation of Innate Immunity to Coronavirus Infection

During virus infection, cells sense viral-pathogen-associated molecular patterns (e.g., nucleic acids) using pattern recognition receptors. For RNA viruses, the main pattern recognition receptors that sense viruses are RIG-I and MDA-5 [97]. These proteins bind the viral dsRNA and signal to activate the transcription factors IRF3 and NF-κB, leading to the production of type I IFNs such as IFN-β. The signaling of IFN-β to neighboring cells induces IFN-stimulated genes (ISGs) and generates an antiviral state that limits virus spread within a tissue.

In a study published in 1993, 20 adult males were experimentally infected with HCoV-229E [98,99]. Disease symptoms were then followed for 6 days. Patients were scored for cold symptoms and the presence of fibrinogen in nasal exudates. There was a significant increase in fibrinogen in patients who developed a cold, suggesting that bulk plasma was exuded across the airway mucosa, indicative of a subepithelial inflammatory response. Since fibrinogen levels peak right before symptoms subsided, the authors suggested that factors in the plasma (immunoglobulins, kinins, complement coagulation, and fibrinolysins) may have contributed to the viral clearance. In a second publication, they also showed increased proinflammatory cytokines, IFN-γ and IL-1β, consistent with responses to viral infections, but not GM-CSF, which is more consistent with TH2 responses [99]. These findings more clearly define an active and typical immune response to facilitate clearance of the virus in otherwise healthy individuals. Many viruses, including HCoVs, have mechanisms of evading both the IFN-induction and response pathways [42,100]. To determine the antiviral response to HCoV-229E and HCoV-OC43 infection, Loo and colleagues measured the activation of both Type I (IFN-β) and Type III (IFN-λ) IFNs [101]. Primary human bronchial epithelial cells were infected and assessed for virus kinetics and immune responses. Consistent with the studies by Akerlund et al. and Linden et al., HCoV-229E elicited strong Type I/III IFN induction and subsequent activation of IFN-stimulated genes (ISGs); however, HCoV-OC43 IFN induction and signaling was indistinguishable from mock infected cells [101]. This suggests that HCoV-OC43 is strongly inhibiting these pathways. Further work will need to determine how the blockade is occurring and if it correlates with what is seen in patients. It remains unclear what impact HCoV-HKU1 has on IFN induction and signaling.

Coronavirus nsp3 is a multifunctional protein that has protease, deubiquitinating (DUB), and deISGylating activity [102,103,104]. The role of HCoV-NL63 PLP has been tested for the ability to disrupt IFN-signaling and has been shown to inhibit IFN-β induction [105]. Surprisingly, this inhibition was independent of protease cleavage activity or DUB activity [105]. The mechanism of this inhibition has not yet been uncovered. HCoV-NL63 PLP also antagonizes STING activity (component of DNA sensing pathways) by likely removing ubiquitination, and thus, blocking STING dimerization in a DUB-dependent manner [106]. While it may seem counterintuitive for an RNA virus to interfere with DNA sensing, the Dengue virus protease also disrupts the cGAS-STING pathways by inhibiting both cGAS and STING [107,108].

Early on during the COVID-19 pandemic, transcriptional analysis revealed highly impaired type I IFN production in patients with severe disease [109]. In these patients, IFN-β mRNA and protein were undetectable and IFN-α was significantly reduced. This reduction in IFN-β was correlated with a downregulation of ISGs. Subsequently, extensive work has gone into determining the viral proteins involved in the suppression of Type I IFNs [109]. Many SARS-CoV-2 proteins have been implicated in the evasion of the Type I IFN pathway. The accessory proteins ORF6 and ORF8 have been shown to block the transcription of IFN-β, ISREs, ISGs, and NK-κB [110]. Additionally, SARS-CoV-2 nsp1 disrupts IFN-induction and responses by either directly mediating the degradation of host transcripts or blocking of translation of host mRNAs [111,112]. IRF3 nuclear translocation is blocked by multiple CoV proteins (Figure 3A) [113,114]. Nsp13 can also disrupt TBK1 activation [114]. Nsp3 uses its deISGylation activity to remove ISG15 from IRF3 to block its activation and subsequent nuclear translocation [115]. Furthermore, it has been shown that the removal of ISG15 from intracellular proteins leads to its secretion and subsequent signaling, which upregulates proinflammatory cytokines that may be responsible for the cytokine storm [116]. SARS-CoV-2 also can dysregulate the IFN-response pathway using nsp1 and nsp13 by blocking phosphorylation of STAT1 and/or STAT2 (Figure 3B) [113]. As these proteins have not been extensively tested in the common cold HCoVs, future studies need to evaluate the role of individual viral proteins on these immune pathways and to examine genetic differences in HCoV-229E that may be responsible for its inability to block the IFN pathways.

### 3.4. Immunopathology and Modulation of Adaptive Immunity to Coronavirus Infections

While, normally, antibodies mediate protective functions, viruses can exploit them for pathogenesis. For SARS-CoV, MERS-CoV, and SARS-CoV-2, severe disease is associated with an overactivation of the immune response and a cytokine storm that leads to lung damage and potentially death. In SARS-CoV, this is due to a delay or impairment in the Type I interferon (IFN) response and an accumulation of proinflammatory monocytes-macrophages [117]. A recent study showed that SARS-CoV-2 is bound by virus-specific antibodies that bind to FcγR to mediate uptake into macrophages by phagocytosis. This leads to abortive infection and inflammatory cell death that ultimately culminates in systemic inflammation that is demonstrated in individuals with severe COVID-19 [118]. Antibodies from individuals with natural SARS-CoV-2 infection promoted infection of monocytes, whereas antibodies from recipients receiving the mRNA vaccination did not. This suggests that IgG antibodies elicited during natural infection are different than the ones elicited by vaccination. The study looked at whether this could be due to afucosylation, which enhances binding to CD16 (an FcγR). While highly afucosylated antibodies mediated increased virus uptake, it was not determined whether there was a difference in afucosylation of antibodies after natural infection compared to vaccination [118]. These findings may explain the decreased severity of COVID-19 in vaccinated individuals.

Neutralizing antibody production is critical for long-term protection from disease. For the common cold HCoVs, broadly reactive antibodies have been detected, but do not provide adequate protection against subsequent infections with either the same HCoV or other HCoVs [119]. This suggests that HCoV infections fail to induce long-lasting, broadly neutralizing antibodies, as evidenced by reinfections with the same or different HCoVs occurring within 6 months. Multiple studies have also demonstrated that reinfection with SARS-CoV-2 occurs and that neutralizing antibodies rapidly wain post natural infection, especially in asymptomatic individuals or those with mild disease [120,121,122,123].

During virus infection, cytotoxic T cells are important for killing virus-infected cells to mediate clearance of the virus [124]. One way that viruses avoid detection by T cells is to target antigen processing and presentation on major histocompatibility complex class I (MHC I) proteins [125]. For instance, SARS-CoV-2 has been shown to disrupt antigen processing and presentation to prevent cytotoxic T cell recognition and killing through the expression of its accessory protein ORF8, which mediates the degradation of MHC I via autophagy and lysosomal breakdown [126].

### 3.5. Seasonality and Endemicity of Human Coronaviruses

The four HCoVs that cause the common cold exhibit seasonality with the peak of cases in the winter and early spring in more temperate climates. In a retrospective study conducted at the Vanderbilt Vaccine Clinic, which looked at samples collected from 1981–2001, the incidence and seasonality of HCoV-229E, HCoV-NL63, and HCoV-OC43 were determined [8]. The incidence of each virus varied by year; however, the peak of disease for each virus remained in the winter and spring months, with the number of peaks and the months in which they occurred variable among the viruses [8]. With the limited time of the SARS-CoV outbreak, seasonality was not established. There has been no correlation of MERS-CoV with a particular season [127]. Early projections have suggested that SARS-CoV-2, post pandemic, will enter a period of seasonality similar to what has been observed for the common cold coronaviruses [128].

One major factor that determines whether a disease is endemic versus an epidemic or pandemic is immunity [129,130]. When SARS-CoV-2 emerged, the population had no prior immunity because it was a novel virus. This resulted in very high levels of infection. The transition from pandemic to endemic relies on building immunity within the population. Since SARS-CoV-2 infection and vaccination does not seem to provide long-lasting protective immunity, the virus will likely transition into an endemic status. As described above, common cold HCoVs cause seasonal disease and lack long-term protection, despite the presence of broadly recognizing antibodies [119]. With increases in vaccination and regular vaccine boosters, the number of cases and severity of disease may become more manageable with time [131].

## 4. Emergence and Evolution of Common Cold Coronaviruses and SARS-CoV-2

### 4.1. Emergence and Evolution of Human Coronaviruses Associated with the Common Cold

The zoonotic origin of the human seasonal coronaviruses remains largely unclear; however, numerous surveillance and ecological studies have provided insights into some likely sources. Of the four common cold strains, the evidence for the proposed zoonotic origin of HCoV-OC43 is most compelling. The data suggests that HCoV-OC43 jumped into humans from a bovine (due to its widely reported similarity to bovine coronavirus, BCoV) or another potential ungulate host around 1890 [132,133,134]. To date, a wide variety of betacoronaviruses have been identified in large ungulates [132]. A respiratory pandemic occurred in humans around the same time and has been proposed as evidence of the emergence event [134,135].

HCoV-229E was the first of the four circulating common cold strains to be identified (in 1966). Since its discovery, 229E-related coronaviruses have been identified in alpacas, camels, and bats [136,137,138,139,140]. Based on differences in spike proteins through potential recombination events, the evidence seems to suggest that the original natural reservoir for HCoV-229E is likely Hipposiderid bats and may have jumped to humans through either a bat or Camelid species [132,141]. The more recently discovered human coronaviruses HCoV-NL63 and HCoV-HKU1 appear to closely correlate based on phylogenetic analyses of known zoonotic coronaviruses with bat and murine coronaviruses, respectively [132]. Studies have demonstrated that bat cells in vitro are capable of sustaining HCoV-NL63 replication [142]. More recently, several bat coronaviruses have been identified, which more closely cluster with the sequences of both HCoV-NL63 and its related human Alphacoronavirus HCoV-229E [143,144,145]. HCoV-HKU1 shares a high sequence identity to several rodent coronaviruses, including the well-studied murine coronavirus model, mouse hepatitis virus (MHV) [7,146].

Coronaviruses have a relatively high mutation rate, which drives intraspecies viral evolution through genetic drift. However, unlike other RNA viruses, coronaviruses also express an exonuclease with RNA proofreading activity, which limits the mutation rate but potentially permits the larger genomes expressed by coronaviruses compared to other RNA viruses [43,147]. Recombination events mediated by the template-switching capacity of the viral RTC appear to also play critical roles in both the evolution and variations in host range observed for some coronaviruses [43,136,148,149]. During mixed infections *in vitro* of closely related coronaviruses, homologous recombination rates of nearly 20–25% have been observed [150,151]. Multiple genotypes or genogroups have been reported for all four human coronaviruses associated with the common cold [122,152,153]. At least eleven different genotypes have been described for HCoV-OC43, the most common circulating common cold coronavirus, with genetic distances between genotypes of at least 0.5% [153]. At least three different genotypes have been reported for HCoV-HKU1, with genetic distances between genotypes ranging from 1.9% to 4.8% [152]. After a fatal case of a coinfection of SARS-CoV-2 and HCoV-229E, genetic analysis of the HCoV-229E isolate revealed a novel genogroup [122]. Retrospective analysis of other historical isolates identified at least five additional genogroups of HCoV-229E. This study also estimated a mean substitution rate for HCoV-229E of 3.03 × 10^−4^ substitutions per site per year, which is consistent with the rates observed for the other human common cold coronaviruses [132].

### 4.2. Emergence and Evolution of SARS-CoV-2

The first documented cases of SARS-CoV-2 occurred in December 2019 in the Wuhan Province of China. Since its emergence, SARS-CoV-2 has undergone extensive evolution while triggering a worldwide pandemic. Significant analysis and discussion have centered around the origin of SARS-CoV-2. All human coronaviruses to date have been traced to a zoonotic origin, and there remains strong compelling evidence to support a similar origin for SARS-CoV-2 [154]. Consistent with the origin of SARS-CoV in 2003, the first cases of SARS-CoV-2 were traced to animal markets which sold live animals (such as civets and raccoon dogs) that were known to be susceptible to SARS-CoV-2 infection [155,156]. In addition to exhibiting high genomic sequence homology to SARS-CoV, SARS-CoV-2 shares very high sequence identity with several bat coronaviruses, which are found in caves throughout China, and high-risk populations near these caves showed serological positivity to SARS-like coronaviruses [157]. Among the bat coronaviruses with the highest sequence homology discovered to date are two strains isolated from Rhinolophus bats: BtCoV-RaTG13, which shares 96.1% identity and was discovered in 2013 in Yunnan, China, and BtCoV-BANAL-52, with 96.8% identity, discovered in Feuang, Laos, in 2022 [60,158]. However, the direct zoonotic link remains unclear, which has led to other possible ideas on the origin of the virus.

Since its emergence, SARS-CoV-2 has undergone rapid evolution with increased detected changes in its spike protein through a combination of both mutations and recombination events [159,160]. Early on, an adaptive D614G mutation occurred in spike, which later led to a near constant stream of additional mutations (including L452R and N501T/Y) in spike impacting the infectivity, ACE2 binding, and pathogenicity of the virus [159,161,162,163]. Despite the variation observed in SARS-CoV-2 throughout the pandemic, several distinct variants of concern (VOC) have been identified. The first primary VOCs were Alpha, Beta, Gamma, and Delta [159]. However, the increased ability of the Delta variant to evade immune responses and exhibit increased transmissibility led to it largely outcompeting the other variants worldwide by middle to late 2021 [164]. In November 2021, a new variant, Omicron, was detected and has since replaced Delta and has resulted in several sublineages, which, at the time of the writing of this review, remain the dominant variants in circulation worldwide [159]. With the recent Omicron BA.1 and BA.2 lineages, mutation rates of 1.4 × 10^−3^ and 1.1 × 10^−3^ substitutions per site per year have been reported, which are greater than what has been observed for any of the human coronaviruses associated with the common cold [165].

### 4.3. Zoonotic Potential for Novel Human Coronaviruses

Coronaviruses mostly infect animals with a wide range of known hosts including mice, cows, birds, bats, camels, horses, and humans. Most known coronaviruses identified to date have been found in bats, which serve as both the primary reservoir and carrier of many coronavirus strains [4]. Although none of the human coronaviruses in circulation today are believed to have originated in humans, there have been documented cases of humans catching these viruses dating back to the 1960s, when HCoV-OC43 and HCoV-229E were discovered [141]. Provided the precedent for zoonotic spread of coronaviruses between humans and other animal hosts, there remains a high potential for future emergent coronaviruses in humans.

There remain several biotic and abiotic factors which may not only promote but accelerate the emergence of novel human coronaviruses. Urbanization and loss of natural habitats increase interactions between humans and potential zoonotic sources of coronaviruses. As evidenced by the proposed origins of HCoV-OC43 (from bovines) and HCoV-HKU1 (from rodents), increases in human population density and interactions with animals will continue to increase the probability of a potential emergence event. In addition, the high current prevalence of human coronaviruses (including SARS-CoV-2 and the co-circulating strains of the common cold) may further increase the probability of favorable recombination events upon zoonotic exposures to other coronaviruses. With the increasing surveillance and discovery of novel bat coronaviruses, more research can be done to study the ecology and biology of these viruses and their zoonotic potential. It has been found that bats are a host for most of the alpha and beta coronavirus strains [166].

## 5. Concluding Remarks

The COVID-19 pandemic has provided clear evidence of the pandemic potential for future emergent coronaviruses. In response to this historic event, increased ecological surveillance and coronavirus sequencing have accelerated our study of coronavirus biology and increased our understanding of the potential for future emergent human coronaviruses. Therefore, it would be worthwhile to continue to investigate how coronaviruses could continue to impact humanity and potential factors that might play a role in a future health crisis.

After evaluating the biology, pathogenesis, and emergence of the human coronaviruses that cause the common cold, we can anticipate that with increased vaccine immunity to SARS-CoV-2, it will become a seasonal, endemic coronavirus that causes less severe disease in most individuals. Much like the common cold CoVs, the potential for severe disease will likely be present in those who lack a protective immune response or are immunocompromised.

## Figures and Tables

**Figure 1 microorganisms-11-00445-f001:**
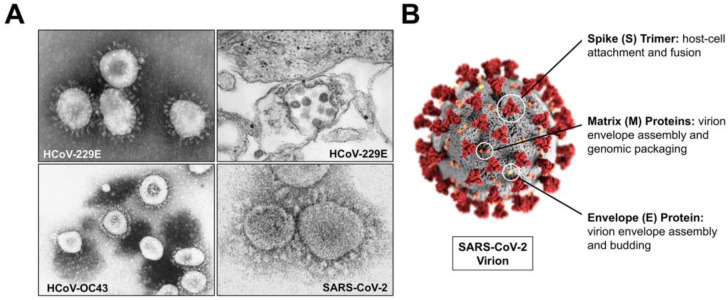
Coronavirus Structure. (**A**) Electron micrographs of HCoV-229E, HCoV-OC43, and SARS-CoV-2. (**B**) Coronavirus virion schematic of SARS-CoV-2 with labeled structural proteins and their respective functions. All images were adapted from the Centers for Disease Control and Prevention (CDC) via the Public Health Image Library (PHIL).

**Figure 2 microorganisms-11-00445-f002:**
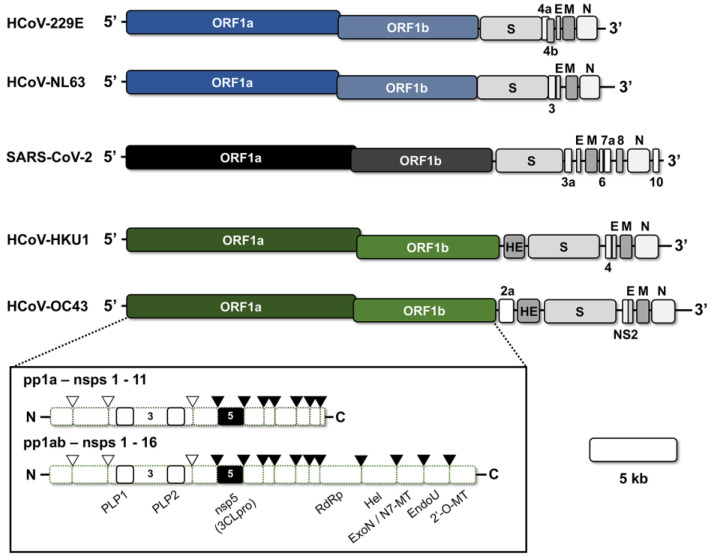
Genomic Organization of the Human Common Cold Coronaviruses and SARS-CoV-2. The genomes of alphacoronaviruses (blue) HCoV-229E and HCoV-NL63 and betacoronaviruses (black, green) SARS-CoV-2, HCoV-HKU1, and HCoV-OC43 are shown with open-reading frames (ORFs) labeled. A 5-kb scale bar is provided. The replicase polyproteins (pp1a and pp1ab) are shown in the box below and the maturation cleavage events mediated by the papain-like proteases (PLPs) and 3CLpro are indicated by the clear or black arrows. E, Envelope; M, Matrix; N, Nucleocapsid; S, Spike; RdRP, RNA-dependent RNA polymerase; Hel, Helicase; ExoN, Exonuclease; N7-MT, N7-methyltransferase; EndoU, Endonuclease; 2′-O-MT, 2′O-methyltransferase.

**Figure 3 microorganisms-11-00445-f003:**
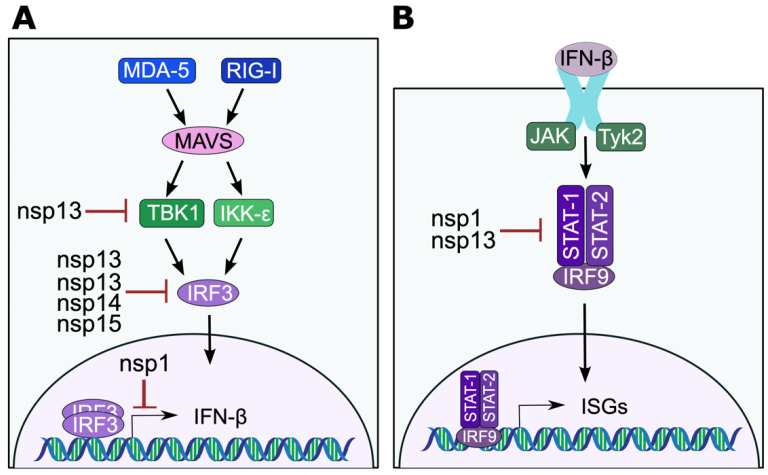
Coronavirus evasion of IFN-induction and response pathways. (**A**) Coronavirus proteins that inhibit IFN-induction targeting TBK-1 or IRF3 protein or transcription of IFN-β. (**B**) Coronavirus proteins that inhibit STAT1/2 in the IFN-response pathway.

## Data Availability

No data were generated for this publication.

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
