# Peer review of "Evaluating the Virology and Evolution of Seasonal Human Coronaviruses Associated with the Common Cold in the COVID-19 Era"

_microorganisms, 2023, doi:10.3390/microorganisms11020445_

Round 1
Reviewer 1 Report
Authors did not mentioned the HCov-NH along with the other 7 HCov
Line 420-421: “undulate animals” maybe authors mean “ungulates”?

Author Response
Didatic/”re-evaluating”: The intention was for the review to be didactic and revisit what we know about the HCoVs that cause the common cold. Then use that knowledge to make an educated guess about the progression of SARS-CoV-2 to becoming an endemic like the common cold HCoVs. We have change re-evaluating to evaluating in the title to make this clearer. We appreciate the reviewer’s comment in bringing this to our attention.
HCoV-NH: We have added a sentence on HCoV-NH and HCoV-NL when HCoV-NL63 is introduced (lines 39-40).
Immunology: As per the request from Reviewer 2, we have included Figure 3 that is a schematic of how CoVs inhibit both the IFN-induction and -response pathways. This figure enhances the pathogenesis section.
Line 420-421: we have updated this typo.
Reviewer 2 Report
This is a well-written and comprehensive review of human CoVs in the face of the COVID-19 pandemic. This is a great manuscript that can be read by the general public.
Author Response
- We appreciated the reviewer’s feedback and have removed Table 1 from the article.
- The section has been updated to reflect only human to human transmission and specific public health measures used to contain the spread of the viruses.
- We agree that a schematic covering how CoV proteins inhibit the IFN-induction and -response pathways would be valuable and have now included this schematic as Figure 3 of the paper.
- We understand the confusion about why the first paragraph was in the adaptive immunity section and have rearranged the paragraph to highlight that it is the virus coopting adaptive immunity (antibodies) to improperly activate innate immune cells.
- We have included a section 5 with concluding remarks.
Reviewer 3 Report
I read with interest the manuscript.
I think authors have created and interesting review with insights into the importance of learning about biological patterns of the virus itself, as well as the zoonotic potential, and human infections. My only suggestion to authors, is to consider a change in the form some information is being presented and to avoid the repetition of information in different sections of the paper.
Here are more precise comments:
1. Table 1 seems unnecessary, since most of this information is already described in section 1.1 and there seems to be only limited distinction with respect to most coronaviruses, except for SARS-CoV-2 which is something that most people already know.
2. Section 3.2 Transmission and cellular infection, has the first 2 paragraphs with repeated information presented in the introduction section. Maybe you could review this section and summarize with only new information that is not previously stated in the introduction section.
3. Sections 3.3 and 3.4 could be summarized in a schema or a table, this would reduce the number of words and create a form to present information in a more didactic form.
4. Section 3.4 can be reviewed, since the first part of the beginning paragraph describes innate mechanisms, not adaptive (Type I INF, monocytes, and macrophages).
5. The paper lacks a conclusion, although some sentences from the last section could be used as part of that conclusion.
Author Response
We thank the reviewer for their kind evaluation.
Round 2
Reviewer 1 Report
Authors properly replied to my concerns.